# Effectiveness of Home-Based Cardiac Rehabilitation, Using Wearable Sensors, as a Multicomponent, Cutting-Edge Intervention: A Systematic Review and Meta-Analysis

**DOI:** 10.3390/jcm11133772

**Published:** 2022-06-29

**Authors:** Varsamo Antoniou, Constantinos H. Davos, Eleni Kapreli, Ladislav Batalik, Demosthenes B. Panagiotakos, Garyfallia Pepera

**Affiliations:** 1Clinical Exercise Physiology and Rehabilitation Laboratory, Department of Physiotherapy, School of Health Sciences, University of Thessaly, 35100 Lamia, Greece; varsamoantoniou@uth.gr (V.A.); ekapreli@uth.gr (E.K.); 2Cardiovascular Research Laboratory, Biomedical Research Foundation, Academy of Athens, 4 Soranou Ephessiou Street, 115 27 Athens, Greece; cdavos@bioacademy.gr; 3Department of Rehabilitation, University Hospital Brno, Jihlavska 20, 62500 Brno, Czech Republic; batalik.ladislav@fnbrno.cz; 4Department of Public Health, Masaryk University Brno, Zerotinovo nam. 617/9, 601 77 Brno, Czech Republic; 5Department of Nutrition and Dietetics, School of Health Sciences and Education, Harokopio University, 176 71 Kallithea, Greece; dbpanag@hua.gr; 6Faculty of Health, University of Canberra, Bruce 2617, Australia

**Keywords:** wearable sensors, home-based cardiac rehabilitation, cardiovascular disease, cardiorespiratory fitness, accelerometer, physical activity

## Abstract

Exercise-based cardiac rehabilitation is a highly recommended intervention towards the advancement of the cardiovascular disease (CVD) patients’ health profile; though with low participation rates. Although home-based cardiac rehabilitation (HBCR) with the use of wearable sensors is proposed as a feasible alternative rehabilitation model, further investigation is needed. This systematic review and meta-analysis aimed to evaluate the effectiveness of wearable sensors-assisted HBCR in improving the CVD patients’ cardiorespiratory fitness (CRF) and health profile. PubMed, Scopus, Cinahl, Cochrane Library, and PsycINFO were searched from 2010 to January 2022, using relevant keywords. A total of 14 randomized controlled trials, written in English, comparing wearable sensors-assisted HBCR to center-based cardiac rehabilitation (CBCR) or usual care (UC), were included. Wearable sensors-assisted HBCR significantly improved CRF when compared to CBCR (Hedges’ g = 0.22, 95% CI 0.06, 0.39; I^2^ = 0%; *p* = 0.01), whilst comparison of HBCR to UC revealed a nonsignificant effect (Hedges’ g = 0.87, 95% CI −0.87, 1.85; I^2^ = 96.41%; *p* = 0.08). Effects on physical activity, quality of life, depression levels, modification of cardiovascular risk factors/laboratory parameters, and adherence were synthesized narratively. No significant differences were noted. Technology tools are growing fast in the cardiac rehabilitation era and promote exercise-based interventions into a more home-based setting. Wearable-assisted HBCR presents the potential to act as an adjunct or an alternative to CBCR.

## 1. Introduction

Cardiovascular diseases (CVD) are a major cause of morbidity and mortality worldwide, thus adding a significant economic burden on national health care systems. Coronary heart disease (CHD) is the most common type of CVD and accounts for a high proportion of all CVD deaths and has more disability-adjusted lifetime than other diseases such as cancer and diabetes [1]. Therefore, secondary prevention interventions that support CVD management are critical in reducing disease burden and health care expenditure. Exercise-based cardiac rehabilitation (EBCR) is highly recommended as the key multicomponent intervention for the prevention of cardiac-induced mortality, the reduction of hospital readmissions, and the improvement of quality of life (QoL) [2]. CR is a safe, efficient, and cost-effective intervention that reduces overall health service expenditure [3]. CR programs are designed to improve physical, psychological, and social functioning by combining medical evaluation, individualized exercise prescription, cardiac risk factor modification, education, and counseling [4,5,6,7,8,9].

Despite the well-documented benefits of CR implementation in the cardiovascular population, attendance rates remain low and suboptimal [10,11]. Accessibility-related factors, including limited availability of programs, unwillingness to participate in group programs, inconvenient timing of programs, career responsibilities, transportation and parking costs, lack of time, and disbelief in their ability to control their CHD, are prominent barriers to CR enrollment and adherence [12,13,14]. Moreover, during the COVID-19 pandemic era, safe distancing measures were adopted to curb the spreading of the virus, thus leading to the temporary cessation of many CR programs, the discontinuation of CR provision, and thus the further deterioration of CVD patients’ cardiovascular function [15,16,17]. Furthermore, a recent systematic review found that CVD patients affected by COVID-19 presented worse outcomes and increased risk of morbidity, whereas COVID-19 itself also induced myocardial injury, arrhythmia, acute coronary syndrome, and venous thromboembolism [18].

The home-based model of CR (HBCR) may act as a sufficient alternative for dealing with some of these barriers and improving cardiac patients’ cardiorespiratory fitness (CRF), QoL, CVD risk factors, mortality, and accessibility/participation rates [19,20,21]. Moreover, the rapid proliferation and widespread use of affordable information and communication technologies (ICTs) in the area of telehealth allow their engagement in the CR procedures, enabling the sufficient provision of feedback, coaching, and specialist consultancy to the CVD population [22]. The significant growth in the use of technology among older adults [23] and the widespread accessibility of the internet also contribute to the implementation of sophisticated telemedicine and mobile CR programs, aiming to better accessibility, individualization, and utilization by cardiac patients. Several systematic reviews have demonstrated the efficacy and feasibility of digital CR interventions in improving cardiac patients’ physical activity (PA) and QoL [24]. Furthermore, patients’ adherence to medical therapy, ability to meet blood pressure and exercise targets, and increased awareness of diet and exercise significance display positive effects in the mobile health CR group [25]. Additionally, a systematic review and meta-analysis has demonstrated that CR telehealth interventions are significantly associated with lower rehospitalization or cardiac events rates and advanced lipid and smoking profiles [22].

Recently, the integration of remote technologies and wearable sensors has enabled the almost real-time monitoring of “at home” patients’ performance data for physical activity features (such as intensity, time, distance traveled, steps taken, and sedentary time), heart rate and blood pressure levels, and cardiac electrical potential waveforms (electrocardiography) can be retrieved through wearable sensors. Subsequently, these recorded data can be assessed almost instantly by the medical staff via remote technology applications, thus allowing constant surveillance and immediate feedback between patients and CR providers. However appealing the concept is, the comprehensiveness of remote CR programs using wearable sensors still lacks proper study investigation. A recent review by Batalik et al. proposes remotely, via wearable sensors, monitored cardiac telerehabilitation, as a feasible, efficient, and safe intervention [26]. Furthermore, cardiac telerehabilitation was demonstrated to be similar in training intensities to conventional outpatient CR in CVD patients with low to moderate cardiovascular risk [27]. 

Though more thorough and systematic search is needed since the integration of wearable sensors in CR procedures is at its early stages and everyday new and more complicated technology is being used, CRF is a powerful and independent predictor of CVD patients’ cardiac and all-cause mortality [7]. Therefore, based on the significance of the CRF, this systematic review aims primarily to explore and examine the effectiveness of wearable sensors-assisted CR in improving CVD patients’ CRF. The secondary aim is to analyze the impact on physical activity (PA), QoL, adherence, and cardiac risk factors compared with center-based CR (CBCR) or usual care.

## 2. Materials and Methods

### 2.1. Study Design

This is a systematic review of randomized controlled trials (RCTs) and is written following the guidelines of the Preferred Reporting Items for Systematic Review and Meta-Analysis (PRISMA). The protocol was registered in PROSPERO (International Prospective Register of Systematic Reviews) (registration number: CRD42021265665) before screening search results, and was conducted according to the PRISMA (Preferred Reporting Items for Systematic Reviews and Meta-Analyses) statement (Appendix A). 

### 2.2. Study Inclusion Criteria

Studies were included if they addressed the implementation of HBCR and encompassed at least two exercise sessions a week. HBCR should be compared to either a usual care group or a CBCR or both. Selected studies should involve interventions among adults (aged ≥18 years and with no restrictions regarding sex, ethnicity, and socioeconomic background), with diagnosed CVD (heart failure, MI, angina, and coronary revascularization), and eligible for phase III of CR. Eligible studies had to involve the assessment of CRF as the primary outcome. Additional inclusion criteria referred to the reporting of at least one more additional outcome measure: PA, QoL, adherence, cardiovascular risk factors, lipid profile, and depression/anxiety levels. The intervention duration should be of at least 8 weeks. The available publications had to be written in English and had to be in full-text version. 

Narrative reviews, preclinical studies, duplicate studies, editorial or opinion articles, grey literature, and conference papers were excluded. Systematic reviews and study protocols were not eligible for inclusion; however, relevant systematic reviews were assessed as a guide and cited where appropriate, and results articles were sought to identify additional RCTs study protocols. 

HBCR interventions were defined as those with at least 50% of the program delivered via ICT, including wireless devices such as sensors, any mobile phone (i.e., feature phone or smartphone), and/or web-based platforms. CBCR is referred to as face-to-face center-based or community-based CR. Usual care was defined as any routine care for CHD, excluding telehealth intervention, without significant ongoing input from a research team.

### 2.3. Search Strategy

A systematic electronic literature search was performed across five electronic databases: PubMed, Scopus, Cinahl, Cochrane Library, and PsycINFO, from 2010 up to January 2022. Systematic searches were conducted by combining the search terms from the four categories of the relevant keywords (i.e., heart disease, program/intervention, mode of delivery, wearable sensors). Keywords are presented in Appendix A. Only full-text articles were included and their reference lists were checked to identify any more potentially eligible studies.

### 2.4. Study Selection Process

Search results were exported to Endnote X9, where, after the exclusion of duplicates, two reviewers (AV, PG) independently screened the titles and abstracts of studies. Those not meeting the eligibility criteria were removed. The full texts of all relevant studies were sought, downloaded, and further evaluated for compliance with the eligibility criteria. Any disagreements between the two reviewers regarding inclusion were resolved by consultation with a third independent reviewer (KE), thus ensuring the minimization of bias, when deciding whether or not to include certain studies. The two reviewers (AV, PG) independently conducted the data extraction from each study. The disagreements were resolved by consulting the previous third independent reviewer (KE).

### 2.5. Data Extraction

Data extraction was performed on the selected studies, including the following domains: author, year, country, sample size, age, and gender of the participants, design, sampling method, description of interventions (mode of delivery, frequency, and duration, and key component), comparator, wearable sensors, outcome measures and time points, results, attrition rate, and handling of the missing data.

### 2.6. Effect Size Measurement

The outcome of interest was the mean difference between the HBCR interventions (CBCR or usual care or both) and the control group from the baseline assessment endpoint in CRF; data were retrieved and recorded by AV, PG that worked independently. Any disagreements were resolved by consensus, or by consultation of a third reviewer (MK). Manuscripts were included in the meta-analysis only if the CRF was adequately reported.

### 2.7. Data Synthesis

Pooled values of weighted mean differences between the HBCR and the CBCR or usual care group, and 95% confidence intervals (CΙ), were calculated using the Der Simonian–Laird random effects, as well as fixed effects models (depending on heterogeneity), using STATA software (version 17, College Station, TX, USA). Estimates of effect size measures were weighted by the inverse of their variances; thus, effect sizes of standardized mean differences were estimated using Hedges’ g statistic and the corresponding 95% CI. The magnitude of Hedges’ g was interpreted as small (g = 0.3), medium (g = 0.5), and large (g = 0.8). Heterogeneity assessed the null hypothesis that all studies evaluated the same effect using the chi-squared test. Inconsistency index (i.e., I^2^) was used to quantify the total variation consistent with inter-study heterogeneity, ranging from 0% to 100%. *p*-values of <0.10 for the chi-square test and I^2^ > 50% were considered to reflect significant heterogeneity [28].

Possible publication bias was assessed using a contour-enhanced funnel plot of each trial’s effect size against the standard error. Funnel plot asymmetry was evaluated by means of the regression-based Egger test for small-study effects. Finally, in the case of multiple assessment time points, the longest one was chosen for inclusion in the meta-analysis.

### 2.8. Risk of Bias (Quality) Assessment

The Cochrane Risk of Bias tool [29] for randomized trials was used to guide the quality assessment of each included study and consists of the following domains: random sequence generation, allocation concealment, blinding of participants and personnel, blinding of outcome assessment, incomplete outcome data, selective reporting, and other bias (e.g., whether study groups were comparable at baseline). Two independent authors (AV, PG) conducted the quality appraisal individually. Any discrepancies were resolved by a third reviewer.

## 3. Results

### 3.1. Study Selection

The initial search from the five electronic databases identified 245 records, of which 52 duplicates were removed. After screening the title and abstracts of 193 records, 144 were excluded for not meeting the inclusion criteria. Forty-nine remaining records were eligible for further full-text review for compliance with the eligibility criteria. The exclusion of 35 records with reasons is documented in Figure 1. Finally, 14 studies were included in this systematic review [30,31,32,33,34,35,36,37,38,39,40,41,42,43].

### 3.2. Risk of Bias of Included Studies

Selection bias related to the generation of the random allocation sequence was considered as low risk, with all trials adequately describing random sequence generation. Five trials reported details [30,31,32,33,40] concerning the sample’s allocation concealment and thus were assessed as low risk. The rest lacked either a detailed description of the procedure or some clarified information and were subsequently classified as unclear.

Considering performance bias, the nature of these trials made the participants or rehabilitation providers’ blinding to group allocation impossible. Nevertheless, in such study designs, the outcome assessors’ blinding to the knowledge of trial allocation can be considered of greater importance. However, only seven studies reported having taken measures to blind outcome assessment [30,31,32,35,36,37,38]. Both attrition and reporting bias domains were mostly rated as having low risk. Only one study reported increased attrition rates and was evaluated as high risk [38]. A summary and a graph of the risk of bias are provided in Figure 2 and Figure 3, respectively.

### 3.3. Study Characteristics

A summary of the characteristics of the included studies is presented in Table 1. Eleven studies were two-arm RCTs [30,31,32,33,37], two were three-arm RCTs [36,41,42], and one included four groups (two intervention and two control) [43] involving a total of 1363 participants (sample size ranging between 28 and 179). Three studies were conducted in Belgium [35,41,42], two in China [30,40], one in the Netherlands [33], one in New Zealand [31], one in Spain [39], one in Iran [43], one in Australia [32], one in Czech Republic [40], one in Poland, one in Germany [36,38], and another one was a multi-center study across Europe [37]. Based on the World Bank database, almost all studies were implemented in countries classified as of high-income, according to their gross national income (GNI) per capita [44]. Sole exceptions were China, classified as an upper-middle-income country, and Iran, which presented a lower-income country classification. Eligible participants in this review were diagnosed with the following: angina [31,40], myocardial infarction (MI) [31,40,41,42,43], acute coronary syndrome (ACS) [33], CHD [35,38,40,41,42], or had undergone coronary revascularization [31,40], ischemic cardiomyopathy (ICM) [39], radiofrequency catheter ablation (RFCA) [39], or chronic heart failure [32,37]. The mean age of participants ranged from 51.4 to 72.4 years and 51.5 to 73.6 years for the intervention and control groups, respectively. A total of 226 females participated in the studies, accounting for 16.6% of the overall sample size. Description of the usual care group varied but mainly referred to encouragement to be physically active, but no participation in supervised CR programs, self-initiated access to CR education sessions, and psychosocial support.

### 3.4. Intervention Characteristics

The CR implementation features (frequency, intensity, duration, and type of exercise) differed significantly among the studies. In particular, four studies reported a 6-month duration of technology-assisted interventions [35,36,37,38,40], seven studies reported a 12-week duration [30,31,32,33,40,41,42], and three studies had an 8-week duration [36,39,43]. The frequency of the exercise sessions ranged from two to six sessions per week, and the duration of each exercise session ranged from 30 min to 80 min per session. Most of the studies reported exercise intensity individually set at 70–80% of each participant’s heart rate reserve (HRR) and 11–13 Borg score of perceived exertion. Only two studies reported lower intensity levels (40–65%) [31,36]. Most programs used individually tailored exercise prescriptions, thus making it difficult to quantify the volume of the exercise taken.

Randomization procedures were reported in all studies. The number and the type of the comparators’ groups differed among the studies. Six studies compared one HBCR group to a traditional CBCR group [31,32,33,34,35,39], five studies compared HBCR to a usual care group [30,36,37,38,40], and one study compared three groups: an HBCR and a CBCR versus a usual care group [42]. One study included four comparators: two intervention and two control groups. Two groups (intervention and control) consisted solely of male participants and two groups (intervention and control) consisted solely of female participants [43].

CBCR programs were based on either a supervised treadmill or cycling exercise, whilst all HBCR programs were orientated to aerobic training. Only two studies based on HBCR interventions included strengthening [38,39] and stretching exercises [30]. Supplementary forms of communication, such as text messages, phone calls, video calls, and emails, were utilized between the participants and the intervention team to provide feedback. Feedback was orientated on adjustment of exercise modalities and features and checking the incidence of adverse events and possible barriers preventing the patients’ participation in the intervention procedures [35,42]. Telephone contacts, delivered once/weekly, were the most common modes of providing behavior change education, psychological support, and evaluation of exercise modalities, training adherence, and CR barriers [33,36,37,40,41]. Hwang et al. provided HBCR patients with educational topics delivered as electronic slide presentations with embedded audio files [32]. Direct messaging via short message service (SMS) or emails, once every week, were also utilized for exercise, dietary, and smoking cessation recommendations [31,35,42]. Support systems with artificial intelligence (AI) were utilized to extract and upload monitored data and provide patients with educational material and motivational and training feedback [38,40].

Stress management and psychological support were the most minor addressed issues mentioned in only four studies [33,36,38,39]. Smoking [35] and dietary [30,35,39] recommendations were also provided whilst only Hwang et al. reported an HBCR being implemented in groups of up to five participants [32]. By design, these trials were impossible to achieve and ensure blinding to group allocation for the participants and the CR professional providers. However, all studies reported measures taken for achieving blinding on the outcome assessment. All studies reported sources of trial funding; though none of them reported funding from any agency with a commercial interest in the results of their study.

### 3.5. Wearable Sensors

Severable wearable sensors were used to assess, monitor, and record vital signs related to the safety of the exercise sessions and the volume of the participants’ PA. Electrocardiographic (ECG) monitoring was used in four studies [30,36,38,39]. Accelerometer data were collected and recorded in four studies toward objectively monitored PA levels [33,35,41,43]. Kraal et al. estimated accelerometer data (ActiGraph wGT3Xþ monitor, Acti-Graph Corp, USA) to determine the HBCR participants’ physical activity energy expenditure (PAEE) and physical activity level (PAL) [33]. HR devices, either chest belts [37,40] or wrist-worn [42], were used to record the exercise data and evaluate training duration and intensity. Additionally, an automatic sphygmomanometer and a finger pulse oximeter were provided to HBCR participants for self-monitoring and verbally reporting their blood pressure, HR, and oxygen saturation levels at the start of each exercise session [32]. A combination of information regarding heart and respiratory rates, single-lead ECG, and accelerometry were provided by a chest-worn wearable sensor (BioHarness 3, Zephyr Technology, Annapolis, MD, USA) [31].

## 4. Primary Outcome

### Cardiorespiratory Fitness

CRF was evaluated as a primary outcome in all the selected studies of this review (Table 2). Almost all included studies determined CRF as the peak oxygen consumption (VO_2peak_) assessed during a maximal cardiopulmonary exercise test (CPET) with respiratory gas analysis on a cycle ergometer (Lode Corrival, Groningen, The Netherlands). In four studies, additional CPET parameters were recorded and used for further assessment, such as the ventilatory anaerobic threshold (VAT) using the V-slope peak heart rate, peak respiratory exchange ratio, both ventilatory thresholds (VT1 (ventilatory anaerobic threshold), VT2 (respiratory compensation point)), HR reserve, oxygen pulse (O2/HR, ml/beat), and aerobic work rate dO2/dW (mL/min/W), VE/VCO2 slope, and VE/VCO2 [38,40,41,42].

Two studies [39,43] carried out an exertion test on a treadmill with continual monitoring with a 12-derivation ECG, using the Bruce protocol. The metabolic equivalent of task (MET), VO_2max_, the total exercise times, and the distance traveled on the treadmill during the exertion test were recorded and used for the evaluation of the participants’ CRF. Additional parameters recorded and used for the evaluation of the physical capacity were the maximum HR reached in the stress test, the HR recovery during the first minute, and the perceived exertion level according to the Borg scale. Only one study performed a six-minute walk test (6MWT) to evaluate the possible effects of its intervention on the physical fitness of its participants [32].

## 5. Secondary Outcomes

### 5.1. Physical Activity

PA was determined as physical activity energy expenditure (PAEE) estimated from data of triaxial accelerometers (Table 2). Steps, sedentary time (duration of sedentary activity at an intensity of ≤1.5 metabolic equivalents of task (METs)), active energy expenditure (PA at an intensity of ≥3 METs), duration of moderate, vigorous PA (≥3 METs), and PA level (PAEE/resting metabolic rate) were parameters recorded and used in the analyses. Cain et al. included the International Physical Activity Questionnaire (IPAQ) for the self-reported PA assessment by their study participants [30], whilst Frederix et al. used both accelerometry data and the IPAQ for PA assessment reasons [35]. On the other hand, Snoek et al. used two questions for the assessment of self-reported physical activity: “How many days per week do you perform moderate to vigorous PA (physical activity)?” and “How many minutes per day do you perform moderate to vigorous PA?” Self-reported habitual physical activity was considered the total number of days per week in which a minimum of 30 min of self-reported moderate to vigorous physical activity was registered [37].

### 5.2. Quality of Life

The Medical Outcome Survey Short Form 36 (SF-36) questionnaire was used to assess the participants’ QoL in four studies, using the corresponding translated version of the SF-36 according to their native language [36,39,41,42]. Kraal et al. used the results from the SF-36 questionnaire to calculate the health utility scores for the cost–utility analyses in their study; whilst the MacNew questionnaire was used for the assessment of the participants’ QoL [33]. Two studies [31,38] used solely the EuroQol five-dimensional (EQ-5D) questionnaire for the evaluation of the QoL, whilst one study [32] combined it with the Minnesota Living with Heart Failure Questionnaire (MLWHFQ).

### 5.3. Training Adherence

Attendance rates were defined either as the number of sessions attended by each participant or as the percentage counted from the total number of accomplished training sessions of an individual participant [30,31,32,33,36,41]. Adherence in the intervention groups was calculated according to the records provided by the wearable sensors (HR zones, accelerometers, ECG recording devices), whilst for the CBCR group, adherence was determined as the number of attended training sessions at the outpatient clinic or patients’ exercise diaries. No adherent was defined that completed <20% of the prescribed number of training sessions.

### 5.4. Cardiovascular Risk Factors/Laboratory Parameters

The evaluation of the main cardiovascular risk factors referred to anthropometric measures such as body mass index (BMI), waist and hip circumference, and biochemical parameters of a fasting blood sample (glucose, total cholesterol, low-density lipoprotein (LDL) cholesterol, high-density lipoprotein (HDL) cholesterol, and triglycerides and glycated hemoglobin (HbA1c)). Evaluation of cardiac risk biomarkers such as CRP and ntBNP were also included in one study [38]. Furthermore, Avila et al. calculated the homeostasis assessment model (HOMA) index using the following formula: fasting plasma glucose (mmol/L) times fasting serum insulin (mU/L) divided by 22.5 [41].

### 5.5. Stress/Patient Satisfaction

Patient satisfaction was measured by the Client Satisfaction Questionnaire (CSQ-8) [32] and the Consumer Quality Index [33]. Psychological status was assessed using the Hospital Anxiety and Depression Scale (HADS) and the patient health questionnaire (PHQ) [32,33]. Cai et al. also used the Health Beliefs Related to Cardiovascular Disease Scale and the Exercise Self-Efficacy Scale to investigate and evaluate the efficacy of their study intervention procedures [30].

### 5.6. Muscle Strength/Balance

Muscle strength was evaluated through the sitting–rising test (SRT), a handgrip strength dynamometer, and quadriceps maximal isometric knee extension strength [32,41]. Balance was measured in one study [32] using the Balance Outcome Measure for Elder Rehabilitation (BOOMER).

## 6. Meta-Analysis

### 6.1. Cardiorespiratory Fitness

A meta-analysis was conducted on pooled data from 12 studies (out of 14 RCT), which compared the HBCR group and the control group (CBCR or usual care or both), after excluding four studies [32,38,39,41] that provided insufficient data for the meta-analysis. The results are displayed in Figure 4 and Figure 5 and in Appendix A.

### 6.2. HBCR versus CBCR

Seven studies investigated the effect of HBCR compared to a CBCR group on participants’ CRF levels. The random and fixed effects model revealed a significant post-intervention between-group difference in favor of the HBCR on CRF with a medium effect size (Hedges’ g = 0.22, 95% CI 0.06 to 0.39), and with low heterogeneity (I^2^ = 0%) (Figure 4). Moreover, the pooled mean difference in favor of the HBCR group on CRF outcome values was 1.27 mL/Kg/min (95% CI 0.24 to 2.30) (Appendix A).

Leave-one-out sensitivity analysis showed that, in general, the overall effect in favor of the HBCR ranged from 0.23 to 0.26. However, when the study by Dehghani et al. was removed, the overall effect dropped to 0.19 (male participants) and to 0.20 (female participants) (Appendix A).

### 6.3. HBCR versus Usual Care

The combined analysis of the five studies evaluating the effects of HBCR to UC group on CRF revealed a nonsignificant effect (Hedges’ g = 0.87, 95% CI −0.87 to 1.85), with a large heterogeneity (I2 = 96.41%; z = 1.75, *p* = 0.08) (Figure 5), although leave-one-out sensitivity analysis showed that with the removal of the Cai et al. study, the overall effect reached significance (Hedges’ g = 0.37, 95% CI 0.18 to 0.56) (Appendix A).

## 7. Other Measurements

### 7.1. Physical Activity

Physical activity behavior was reported either through the use of accelerometers (Table 2) [33,35,41,42] assessing steps per day, sedentary time, and daily minutes of moderate, vigorous PA, or through self-reported days per week of moderate-vigorous PA, International Physical Activity Questionnaire [30,35], and exercise habits (number of participants reporting 30 min of moderate activity performed 3–5 times/week) [37]. No interaction effect was found for PA levels in all assessment endpoints for studies extracting PA data from accelerometers [33,41,42]. On the contrary, self-reported PA demonstrated improvements in PA levels within telerehabilitation groups compared to control groups [30,35,37].

### 7.2. Quality of Life

The results of the Medical Outcome Survey Short Form (SF-36) questionnaire, used to measure the QoL, revealed different results (Table 2). In the study of Bravo-Escobar et al., the only difference between the study groups was that the QoL scores were significantly higher in the CBCR group [39]. On the contrary were the results of another study, where the QoL was improved significantly in the intervention group (*p* < 0.001) [36]. In addition, other studies showed that total QoL improved significantly in both groups (*p* < 0.01), but no significant difference was found between groups [40] or in the overall score for QoL [41,42]. The Minnesota Living with Heart Failure Questionnaire (MLWHFQ) [32] and the EuroQol five-dimensional (EQ-5D) [31,32,38] did not report any between-group differences regarding their QoL.

### 7.3. Cardiovascular Risk Factors/Laboratory Parameters

The effects of exercise-based interventions on the CVD patients’ risk profiles reveal controversial results (Table 2). The glycemic control remained stable in all study groups [35] or increased in the control group and remained stable in the intervention group [37]. A tendency towards higher total cholesterol and low-density lipoprotein values was observed in all study participants [31,35,41]. Diastolic blood pressure remained stable in the intervention groups, increasing in the control group [37,41]. Maddison et al. reported smaller waist (*p* = 0.04) and hip circumferences (*p* = 0.04) outcomes during the intervention period in the intervention group, though this became absent at the follow up after the 24 weeks (no intervention) period [31].

## 8. Discussion

This current systematic review and meta-analysis of the available information has identified a positive effect of the wearable-sensors-assisted HBCR with improvements in patients’ CRF, whether the HBCR was used as an adjunct or as an alternative to CBCR. This finding is in accordance with previous systematic reviews that also proclaim the feasibility and effectiveness of digital HBCR in improving the patients’ CRF levels [7,24,45,46,47]. Additionally, the participants’ adherence rates appear to be higher in the intervention HBCR groups (Table 2), thus promoting a more profound aerobic training and probably explaining the more beneficial impact of HBCR on CRF levels when compared to the CBCR group outcomes. Surprisingly, no significant differences in symptom-limited exercising testing between HBCR and UC groups were observed. Similar results were presented in a recent meta-analysis where the HBCR group’s CRF did not differ from the UC in CPET results [46], though when Cai et al.’s study [30] was omitted from the meta-analysis, CRF outcome values differed significantly in favor of the HBCR group. The usual care group was encouraged to participate in an out-of-hospital, unsupervised exercise aerobic training, though with specific intensity prescriptions. Heterogeneity results reveal a contravention of Cai et al.’s study to the rest of the usual care groups, to which only standard counseling to remain physically active was given effect (Appendix A).

In this systematic review, objective recording and evaluation of PA activity, via wearable sensors, revealed an inability of HBCR interventions to engage cardiac patients in a more active lifestyle. A previous meta-analysis of eHealth CR interventions, though, showed significant improvement in PA outcomes, in favor of the intervention groups [48], thus leading to inconsistency compared to our study results. This inconsistency may be explained because most of the studies included in this review based their PA evaluation on objective data monitoring and recording via accelerometers. Objective PA monitoring may have prevented a personal, subjective determination of physical status that could have led to a potential measurement recall bias [46,49]. In addition, intervention patients may have presented increased sedentary time levels, due to their engagement in the regular, programmed exercise sessions, thus making them more reluctant to seek additional physical exercise training.

Furthermore, wearable sensors with an accelerometer and ECG, combining AI algorithms and continuous monitoring, enabled the more accurate detection and identification of patients’ PA. Using AI, online platforms can facilitate remote communication between patients and clinicians, thus allowing consultations and prescription adjustments. Alternative chatting methods, such as e-mailing within a website, could also promote unlimited communication between patients and rehabilitation teams [49]. AI could improve the efficacy and effectiveness of HBCR by advancing its comprehensiveness; thus, further search on the utilization of AI is highly recommended.

Although the patients’ QoL was assessed in almost all included studies in this review, the majority of them reported no significant between-group differences. Psychological parameters, such as anxiety and depression, were evaluated by only three studies in this review and the implementation of wearable-sensors-assisted CR interventions showed no significant effects [33,37,38]. These findings are supported by a recent meta-analysis that revealed comparable effectiveness in psychological outcomes between technology-assisted HBCR and CBCR [50]. Moreover, in this review, adherence rates were comparable between HBCR and CBCR, with two studies reporting favorable effects in participation percentages for the intervention groups [30,32]. Thus, there is an indication that HBCR may have the potential to act as an alternative to overcome the barriers preventing the patients’ participation. Especially if the HBCR patients receive appropriate monitoring and constant guidance/feedback, their adherence rates appear to be higher than the ones attending CBCR [32,33,40,42].

A systematic review and meta-analysis demonstrated a reduction in cardiovascular risk factors (blood pressure, lipid profile, and smoking status) at medium- to long-term follow-up compared to comparison groups [22]. Contrarily, our review showed no significant differences between intervention and control groups on modifiable cardiac risk factors. Similarly, Chong et al. indicated that technology-assisted HBCR demonstrated comparable results to CBCR [50]. This discrepancy in our findings may be explained by the substantial improvements in cardiac risk profile derived from the major advancements in diagnostic and therapeutic procedures concerning CVDs, the systematic use of cardio-protective pharmacotherapy, and the adoption of the Mediterranean-type diet as a protective tool against recurrent cardiac events [51].

This systematic review provides a deep insight into the feasibility of implementing digital HBCR as an alternative method to widen access to CR for most of the population suffering from heart diseases. Moreover, this review stands out from previous ones since it emphasizes the understanding of the efficiency of digital HBCR interventions, incorporating the use of wearable sensors for the telemonitoring and the tele-guidance of the exercise sessions in the participants’ home environment. The implementation of wearable sensors-assisted HBCR programs can possibly act as a tool to overcome the several barriers that prevent the cardiac patients’ participation in CR interventions. Real time monitoring, through wearable sensors technology, could allow clinicians to implement CR programs even during pandemic eras and address the cardiac population that is lacking economic background or lives in rural, isolated locations. The wearable sensors-assisted HBCR could play the role of an “adjunct” or a “substitute” to conventional CBCR, based on each cardiac patient’s personal needs and current socioeconomic circumstances. Furthermore, there is a profound need for the scientists and the technology engineers to continuously improve and update the standards and the provided functions of the wearable sensors that can assist the implementation of the CR programs in the cardiac patients’ home settings.

## 9. Limitations

Although this systematic review is probably the first that investigates and evaluates the effectiveness of HBCR interventions incorporating wearable sensors, it displays several limitations. Small sample size and composition are some of them, since many studies had fewer than 100 participants and most of the patients in the included studies were males. Moreover, information about the socioeconomic background, educational level, or place of residence was missing from the participants’ demographic characteristics data, though such information may be necessary for a better understanding and interpretation of the findings of the studies, since they may play a role in the effectiveness and the efficacy of the CR interventions. Furthermore, most of the studies were conducted in high-income countries, thus, limiting the potential to generalize the results in countries with lower socioeconomic profiles. Additionally, the included studies were limited to English-written papers and only original published research articles, which might lead to missing other relevant literature in other languages or information available from the grey literature.

## 10. Conclusions

Overall, we have demonstrated that HBCR interventions using wearable sensors can be as effective as CBCR. If their implementation is achieved on a larger scale, HBCR with wearable sensors has the potential to increase the accessibility, adherence, and participation rates in CR interventions, by helping to overcome several barriers that prevent CR participation. Rural population, pandemic circumstances (such as the COVID-19 pandemic), and female cardiac patients’ evolvement in CR are aspects that need to be taken under consideration in further studies. Continuous technology advancement of the wearable sensors will help integrate them more successfully into the CR procedures.

## Figures and Tables

**Figure 1 jcm-11-03772-f001:**
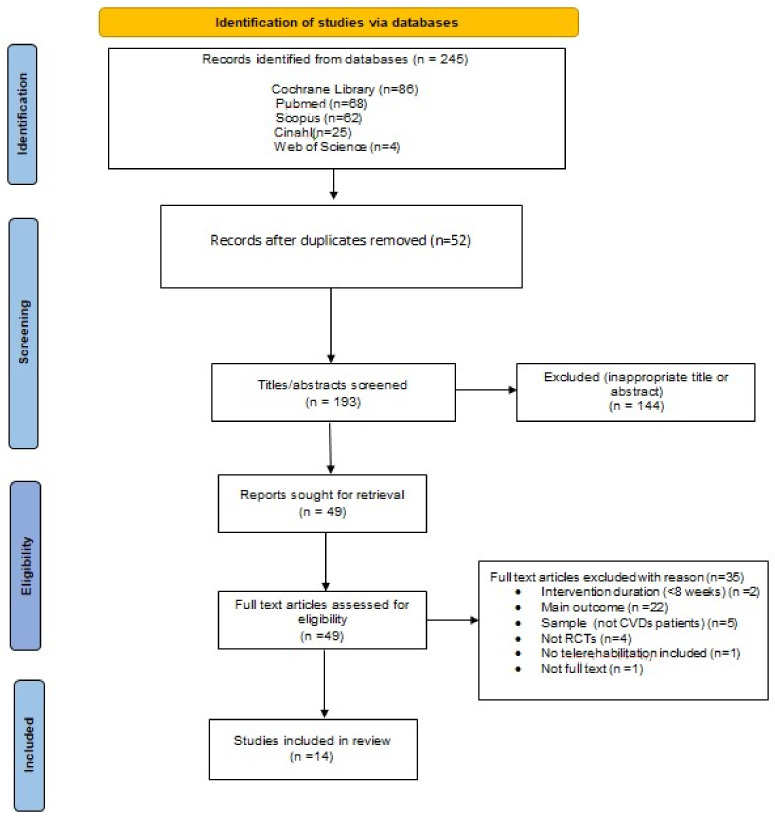
Flowchart of the study.

**Figure 2 jcm-11-03772-f002:**
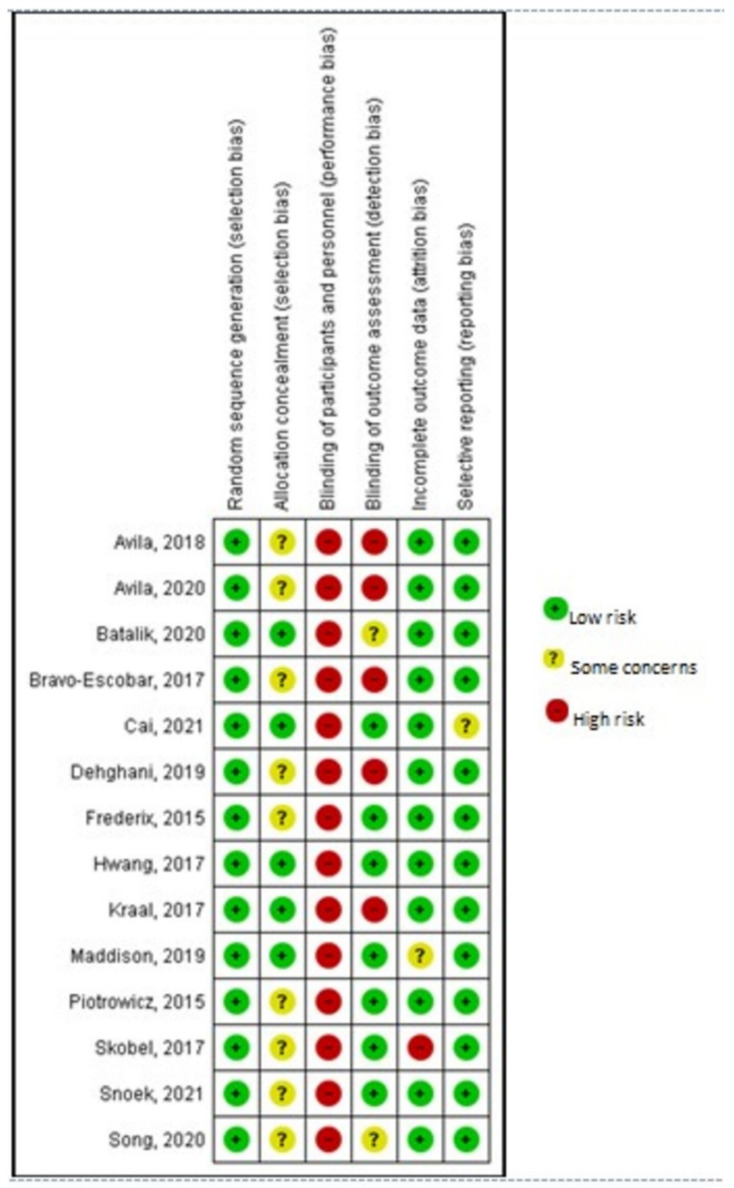
Risk of bias summary. Avila, 2018 [42]; Avila, 2020 [41]; Batalik, 2020 [40]; Bravo-Escobar, 2017 [38]; Cai, 2021 [30]; Dehghani, 2019 [43]; Frederix, 2015 [34]; Hwang, 2017 [32]; Kraal, 2017 [33]; Maddison, 2019 [31]; Piotrowicz, 2015 [35]; Skobel, 2017 [37]; Snoek, 2021 [36]; Song, 2020 [39].

**Figure 3 jcm-11-03772-f003:**
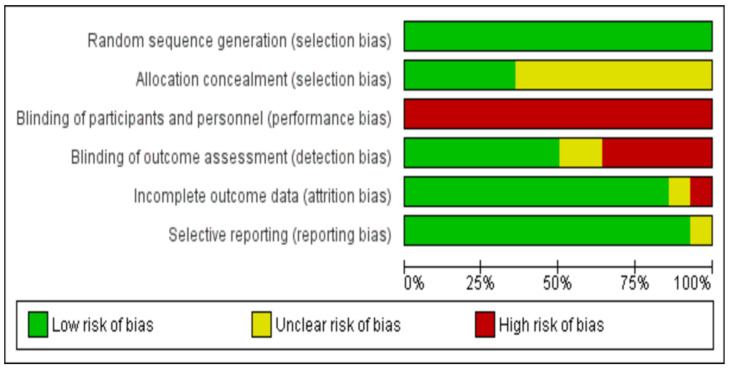
Risk of bias graph.

**Figure 4 jcm-11-03772-f004:**
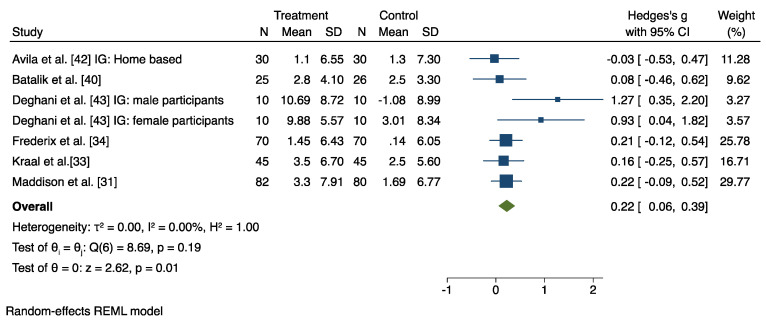
Results from the restricted maximum likelihood (REML) random effects meta-analysis (Hedges’ g criteria), concerning the difference in cardiorespiratory fitness (CRF) change post-intervention, between the home-based cardiac rehabilitation group (HBCR) and the center-based rehabilitation group (CBCR) [31,33,34,40,42,43].

**Figure 5 jcm-11-03772-f005:**
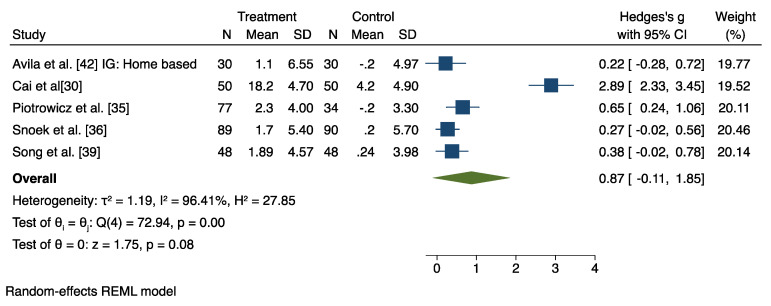
Results from the restricted maximum likelihood (REML) random effects meta-analysis (Hedges’ criteria), concerning the difference in cardiorespiratory fitness (CRF) change post-intervention, between the home-based cardiac rehabilitation group (HBCR) and the usual care group (UC) [30,35,36,39,42].

**Table 1 jcm-11-03772-t001:** Study characteristics.

Author (Year) Country	Study Design	Population (P): a. Number of Participants (*n*) b. Diagnosis c. Age (Mean ±SD) d. Female, *n* (%)	Intervention (I): a. Number (*n*) b. Duration/Frequency (Per Week) c. Intervention Outline d. PA Prescription	Control (C): a. Number (*n*) b. Outline	Wearable Sensors	Outcome (O): a. Primary b. Secondary	Remarks: a. Attrition b. ITT c. MDM d. Protocol e. Funding
Avila et al. (2018)/Belgium [42]	Three-arm parallel RCT	a. *n* = 90 b. CAD, previous MI c. Sample: 61.2 ± 7.6 HB-CRG: 58.6 ± 13 CB-CRG: 61.9 ± 7.3 CG: 61.7 ± 7.7 d. HB-CRG: 4 (13) CB-CRG: 3 (10) CG: 3 (10)	HB-CRG a. *n* = 30 b. 12 weeks/6–7 days per week c. 3 supervised sessions for individualized exercise prescription before the intervention, use of the sensors and data uploading procedures. Weekly feedback via phone or email d. at least 150 min of exercise/week at 70–80% of HRR. CB-CRG a. *n* = 30 b. 12 weeks/3 sessions per week c. 3 exercise sessions at an outpatient clinic d. ~150 min of endurance training (2 × 7 min of cycling, 2 × 7 min of treadmill walking/running, 7 min of arm ergometry or rowing, and 2 × 7 min of dynamic calisthenics) and relaxation. Exercise load adjusted to target HR (70–80% of the HRR).	a. *n* = 30 b. CG: usual care (counseling to remain physically active).	HR monitor (Garmin Forerunner 210, Wichita USA) Accelerometer Sensewear Mini Armband (BodyMedia, Inc., Pittsburgh, PA, USA).	a. Cardiorespiratory fitness symptom-limited CPET (VO_2_max). b. PA, lipid profile, muscle strength and endurance, HOMA index.	a. HB-CRG:2 CG:4 b. Yes c. Yes d. NR e. Yes
Avila et al. (2020)/Belgium [41]	Three-arm parallel RCT	a. *n* = 80 b. CAD, previous MI c. HB-CRG: 62.2 ± 7.1 CB-CRG: 62.0 ± 7.4 CG: 63.7 ±7.4 d. HB-CRG: 3 (12%) CB-CRG: 3 (10%) CG: 2 (08%)	a. *n* (HB-CRG):26 *n* (CB-CRG): 29 b. 9-month follow-up of Avila et al. (2018) study. Solely counseling to remain physically active to all study groups. Accelerometer use for a minimum of five consecutive days.	a. *n* (CG): 25 b. CG: usual care (counseling to remain physically active).	Accelerometer Sensewear Mini Armband (BodyMedia, Inc., Pittsburgh, PA, USA).	a. Cardiorespiratory fitness symptom-limited CPET (VO_2_max). b. PA, lipid profile, muscle function, QoL.	a. HB-CRG:4 CB-CR:1 CG:5 b. No c. No d. NR e. Yes
Batalik et al. (2020)/Czech Republic [40]	Single prospective RCT	a. *n* = 56 b. CVD (MI, angina, MI, CRV) c. ITG: 56.5 ± 6.9 ROT: 57.7 ± 7.6 d. ITG: 4 (15%) ROT: 5 (20%)	a. *n* = 28 b. 12 weeks/3 sessions per week c. 2 supervised training sessions in the outpatient clinic before home intervention. Once/a week, feedback provided d. 3 sessions/week of 10′ warm-up, 60′ aerobic phase (walking or cycling) at moderate rate (70–80% of HRR) and 10′ cool-down.	a. *n* = 28 b. 12 weeks/3 sessions per week. Supervised exercise workout in an outpatient clinic (10′ warm-up, 60′ aerobic phase (cycling on ergometers and walking on treadmill)/week at 70–80% of HRR) and 10 min cool-down).	Wrist HR monitor M430 (Polar, Kempele, Finland).	a. Physical fitness symptom-limited CPET (VO_2_max). b. QoL, adherence.	a. ITG:2 ROT:3 b. No c. No d. Yes e. Yes
Bravo-Escobar et al. (2017)/Spain [38]	Multicenter RCT	a. *n* = 28 b. CAD (ICM with CRV) c. Hospital: 55.64 ± 11.35 Home: 56.50 ± 6.01 d. Hospital: 0 (0%) Home: 0 (0%)	a. *n* = 14 b. 2 months/3 sessions per week c. supervised exercise session in the CR unit once a week combined with an 1 h home walking program for at least two more days a week. Once a week strength-training and health education session at the hospital and group psychotherapy. d. 3 sessions of 1 h at 70% (1st month) and 80% (2nd month)of the HRR	a. *n* = 14 b. 2 months/3 exercise sessions per week of 1 h at 70% (1st month) and 80% (2nd month) of the HRR in an outpatient clinic, counseling for further exercising at home. Once a week: strength-training, health education session at the hospital and group psychotherapy.	Remote ECG monitoring device NUUBO^®.^	a. Exercise capacity (exertion test), SBP, DBP, lipid profile, QoL, adverse events.	a. Hospital: 0 Home: 1 b. No c. No d. No e. Yes
Cai et al. (2021)/China [30]	Single-center, prospective RCT	a. *n* = 100 b. RFCA c. IG:57 ± 11 CG: 57 ± 9 d. IG: 18 (36.7%) CG: 16 (33.3%)	a. *n* = 50 b. 12 weeks/5 times per week c. aerobic training. Mobile application-guidance and device telemonitoring. d. 5 sessions of 65 min each at HR _target_ and HR _alarm._	a. *n* = 50 b. 12 weeks/5 sessions of 65 min each at HR _target_ and HR _alarm_ per week: standard treatment, aerobic training.	ShuKang app (Recovery Plus Inc., China). Portable ECG recording device.	a. Physical fitness symptom-limited CPET (VO_2_max). b. PA (IPAQ), adherence, health beliefs, self-efficacy.	a. IG:1 CG:2 b. No c. No d. Yes e. No
Dehghani et al. (2019)/Iran [43]		a. *n* = 40 b. MI c. MIG: 51.4 ± 7.97 MCG:51.1 ± 7.86 FIG: 51.5 ± 6.96 FCG: 53 ± 7.33 d. 20 (50%)	a. *n* = 10 (male), *n* = 10 (female) b. 8 weeks/5 times per week. c. 2 IGs: male/female participants. Walking exercise program with step counter feedback. d. 5 sessions of 45′–60′ duration (7′ warm-up, 40′ walking, 7′ recovery and stretching exercises) at the 11–13 Borg scale. 10% increase in number of steps/week	a. *n* = 10 (male), *n* = 10 (female). b. 8 weeks/5 sessions of 45′-60′ duration (7′ warm-up, 40′ walking, 7′ recovery and stretching exercises) at the 11–13 Borg scale. 10% per week: walking exercise program without step counter feedback.	NR	a. Functional capacity (treadmill test): METs, VO_2_max, total time, HRmax and distance travelled during treadmill testing.	a. IGs:0 CGs:0 b. No c. No d. NR e. Yes
Frederix et al. (2015)/Belgium [34]	Multicenter, prospective RCT	a. *n* = 140 b. CR patients c. IG: 61 ± 9 CG: 61 ± 8 d. IG: 10 (14%) CG: 15 (21%)	a. *n* = 70 b. 24 weeks/2 times per week c. 12 weeks CBCR and 24 weeks telerehabilitation program (starting from the 6th week of the CBCR). Aerobic training, dietary/smoking cessation/PA guidance. Feedback once weekly (email/SMS). d. 2 sessions of 45′–60′/session at HR _target_ and/or workload, of an intensity at VO_2_max (as achieved in baseline CPET) and calculated BMI.	a. *n* = 70 b. 12 weeks/2 sessions of 45′–60′/session at HR_target_ and/or workload of an intensity between their VT1 and RCP: endurance training (walking/running and/or cycling and arm cranking). Consultation with dietician and psychologist at the rehabilitation center.	Yorbody accelerometer Belgium.	a. Vo_2_max(CPET). b. PA (accelerometer, IPAQ), lipid profile, HbA1c, QoL.	a. IG:1 CG:0 b. Yes, except 1 (non CVD pathology) c. No d. Yes e. Yes
Hwang et al. (2017)/Australia [32]	Two-group, parallel, non-inferiority RCT	a. *n* = 53 b. Chronic HF) c. IG: 68 ± 14 CG: 67 ± 11 d. IG: 5 (21%) CG: 8 (28%)	a. *n* = 24 b. 12 weeks/2 times per week. c. Group-based telerehabilitation with real time. Report of BP, HR and oxygen saturation levels and 15′ educational interactions at the start of each exercise session. d. 60′ of exercise/session (10′ warm-up, 40′ aerobic and strength exercises, and 10′ cool-down). Exercise intensity commenced at 9 (very light) and gradually progressed towards 13 (somewhat hard) on Borg scale.	a. *n* = 29 b. 12 weeks/2 exercise sessions of 60′ per week: aerobic training, education sessions at the hospital. Exercise intensity commenced at 9 (very light) and gradually progressed towards 13 (somewhat hard) on Borg scale.	Automatic sphygmanometer, finger pulse oximeter.	a. Functional capacity (6 MWT). b. Balance tests (BOOMER), 10MWT, strength (grip, quadriceps), urinary incontinence, quality of life (MLWHFQ, EQ-5D), patient satisfaction(CSQ-8), attendance rates, adverse events.	a. IG:1 CG:3 b. NR c. NR d. Yes e. Yes
Kraal et al. (2017)/Netherlands [33]	Prospective RCT	a. *n* = 90 b. CR patients after ACS or PCI or CABG c. IG: 60.5 ± 8.8 CG: 57.7 ± 8.7 d. IG: 5 (11%) CG: 5 (11%)	a. *n* = 45 b. 12 weeks/at least two training sessions a week c. 3 supervised training sessions in the outpatient clinic. Once a week telephone feedback on training modalities. Motivational Interviewing. d. 2 sessions of 45–60 min each at an intensity of 70–85% of the HR_max_ as assessed during the CPET at baseline	a. *n* = 45 b. 12 weeks/2 group-based, supervised training sessions (cycle ergometer, treadmill) of 45′–60′each at an intensity of 70–85% of the HR_max_ as assessed during the baseline assessment in the outpatient clinic.	HR monitor (Garmin FR70) Triaxial accelerometer (ActiGraph wGT3Xþ monitor).	a. PeakVO_2_ (CPET), PA (PAEE, PAL). b. QoL(SF-36), patient satisfaction (Consumer Quality Index), psychosocial status (HADS, PHQ), training adherence and cost effectiveness.	a. IG:8 CG:4 b. yes c. NR d. yes e. yes
Maddison et al. (2019)/New Zealand [31]	Two-arm RCT	a. *n* = 162 b. CHD (MI, angina, CRV) c. IG: 61.0 ± 13.2 CG: 61.5 ± 12.2 d. IG: 13 (15.9%) CG: 10 (12.5%)	a. *n* = 82 b. 12 weeks/3 week c. monitored exercise and remote real –time coaching provision on REMOTE-CR platform Theory-based education content delivered via SMS. d. 3 exercise sessions/week and encouragement to be active >5 days/week of 30′ to 60′ at an intensity of 40–65% HRR.	a. *n* = 80. b. Supervised exercise sessions in CR clinics.	Wearable sensor (BioHarness 3, Zephyr Technology, USA): HR and respiratory rates, single lead ECG and accelerometry Accelerometer Actigraph (GT1M, ActiGraph Corp, USA).	a. Symptom-limited CPET (VO_2_max). b. PA, SBP/DBP, BMI, lipid profile, BG, QoL(EQ-5D), cost effectiveness.	a. IG:17 CG:11 b. No c. Yes d. Yes e. Yes
Piotrowicz et al. (2015)/Poland [33]	Single-center, prospective, parallel-group RCT	a. *n* = 111 b. HF c. IG: 54.4 ± 10.9 CG: 62.1 ± 12.5 d. IG: 11 (15%) CG: 1 (3%)	a. *n* = 77 b. 8 weeks/5 times per week c. 3–6 monitored exercise training sessions before the intervention. Telemonitored and telesupervised Nordic Walking(NW) with the use of EHO mini device (electrocardiogram data). Psychological support via telephone. d. 5 sessions of 5′–10′ warm-up (breathing and light resistance exercises, calisthenics), a 15′–45′ NW training, and a 5′ cool-down. Training intensity set according to RPE and the training HR range (40–70% HRR).	a. *n* = 34. b. No guided exercise training. Only consultation for suitable lifestyle changes and self-management according to guidelines.	EHO mini device—ECG data recorder (Pro Plus Company, Poland).	a. Functional capacity—VO_2_max (CPET). b. Effectiveness of rehabilitation (workload duration in CPET, 6MWT distance, QoL), safety, adherence, acceptance of telemonitoring.	a. IG:2 CG:1 b. NR c. NR d. NR e. Yes
Skobel et al. (2017)/Germany [37]	A prospective, international, multi-center RCT	a. *n* = 118 b. CAD referred for CR c. IG: 60 ± 50.65 CG: 58 ± 52.67 d. IG: 5 (9%) CG: 8 (13%)	a. *n* = 55 b. 6 months/NR c. Training under guidance of the GEx system. Exercise prescriptions continuously reviewed and adjusted as needed. d. Endurance training (cycling, walking) and resistance training at a predefined HR_target_ zone.	a. *n* = 63. b. 6 months/NR, report of daily physical activities on a paper dairy.	GEX system: info on medical profile, educational material and motivational feedback, sensor for acquisition of vital signs for immediate feedback with respect to training intensity.	a. Physical capacity (CPET). b. Compliance, fear, anxiety (HADS), QoL(EQ-5D), BP, EF, LDL.	a. IG:36 CG:21 b. No c. No d. NR e. Yes
Snoek et al. (2021)/Netherlands [36]	Multicenter, parallel RCT	a. *n* = 179 b. HF 54.4 ± 10.9 c. IG: 72.4 ± 5.4 CG: 73.6 ± 5.5 d. IG: 20 (22%) CG: 14 (16%)	a. *n* = 89 b. 6 months/5 days per week. c. HBCR exercise training. Use of smartphone application to capture training modalities. Motivational interviewing applied by telephone: weekly in the 1st month, every other week in the 2nd month, and monthly until completion. d. 5 sessions of 30′ moderate intensity exercise training.	a. *n* = 90. b. No provision of CR, only standard care.	MobiHealth BV smartphone application. HR belt.	a. Physical fitness: VO_2_peak (CPET). b. PA, lipid profile, HbA1c, adverse events, QoL(SF-36v2), depression (PHQ-9) mortality, hospitalization.	a. IG:6 CG:2 b. Yes c. Yes d. Yes e. Yes
Song et al. (2020)/China [39]	Two-arm RCT	a. *n* = 106 b. Stable CHD c. IG: 54.17 ± 8.76 CG: 54.83 ± 9.13 d. IG: 5 (10.4%) CG: 8 (16.7%)	a. *n* = 53 b. 6 months/3–5 times per week. c. Telemonitored HR during PA. Feedback on patients’ exercise frequency/intensity, BP, and HR before and after exercise. Feedback via SMS and telephone call. d. 3–5 sessions of 30′ at an intensity set at HR at aerobic threshold.	a. *n* = 53. b. Usual care (routine discharge education and outpatient follow-up).	HR belts (Suunto).	a. Exercise tolerance-symptom-limited CPET (VO_2_peak). b. SBP/DBP, lipid profile.	a. IG:5 CG:5 b. NR c. NR d. Yes e. Yes

6 MWT, 6 min walk test; ACS, acute coronary syndrome; BG, blood glucose; BMI, body mass index; BP, blood pressure; CABG, coronary artery bypass graft; CAD, coronary artery disease; CBCR, center-based cardiac rehabilitation; CG, control group; CHD, coronary heart disease; CPET, cardiopulmonary exercise testing; CR, cardiac rehabilitation; CRV, coronary revascularization; CSQ-8, client satisfaction questionnaire; ECG: electrocardiograph; EF, ejection fraction; DBP, diastolic blood pressure; FIG: female intervention group; FCG: female control group; HADS, hospital anxiety and depression scale; HbA1c, hemoglobin A1c; HBCTR, home-based cardiac telerehabilitation; HDL, high-density lipoprotein; HF, heart failure; HR, heart rate; HRR, heart rate reserve; ICM, ischemic cardiomyopathy; IG, intervention group; IPAQ, international physical activity questionnaire; ITT, intention-to-treat; LDL, low-density lipoprotein; MDM, missing data management; MI, myocardial infarction; MIG: male intervention group; MCG: male control group; MLWHFQ, Minnesota living with heart failure questionnaire; NR, not reported; PA, physical activity; PAEE, physical activity energy expenditure; PCI, percutaneous coronary intervention; PHQ, patient health questionnaire; QoL, quality of life; RCT, randomized controlled trial; RFCA, radio frequency catheter ablation; REMOTE-CR, remotely monitored exercise-based cardiac rehabilitation; RCP, respiratory compensation point; SBP, systolic blood pressure; SD, standard deviation; SMART-CR/SP, smartphone-based-cardiac rehabilitation/secondary prevention; VT1, first ventilatory threshold; VE/VCO_2_, carbon dioxide equivalent; vCRP, virtual cardiac rehabilitation program; VO_2_, oxygen consumption.

**Table 2 jcm-11-03772-t002:** Results reported in studies.

Author/Year	Baseline/Follow Up at ….	Primary Measure/Outcome Values: From Baseline at Follow Up	Secondary Measures/Outcome Values: From Baseline at Follow Up
Avila et al. (2018) [42]	6 months	Cardiorespiratory fitness VO_2_peak (mL/kg^−1^/min^−1^) Improved in home-based group: from 26.7 (6.55) at 27.8 (6.83) and center-based group: from 25.4 (7.32) at 26.7 (7.90). VT1 (mL/kg^−1^/min^−1^) Improved in home–based group: from 19.5 (1.07) at 21.5 (1.07) and center-based group: from 19.5 (1.04) at 20.4 (1.04). VT2 (mL/kg^−1^/min^−1^) Improved in home–based group: from 24.9 (5.25) at 26.3 (6.98) and center-based group: from 22.7 (6.95) at 24.2 (7.13).	Physical activity Remained constant after the intervention (P-time = 0.73). Significant increase in sedentary time in the center-based group (P-interaction = 0.02). Significant correlation of VO_2_peak with PA duration (ρ = 0.53; *p* < 0.001) and active energy expenditure (ρ = 0.37; *p* < 0.001). Strength/Endurance >Stable isometric handgrip/quadriceps strength (HG), and endurance. Cardiovascular risk factors No change. Only an increase in HOMA index (P-time = 0.05). Quality of Life No significant changes in the overall score for QoL. (P-interaction = 0.57), the physical (P-interaction = 0.50) and mental (P-interaction = 0.85) composite scores. Adherence HBCR: 2.5 sessions/week (range: 12–60 sessions for 12 weeks). CBCR: 2.0 sessions/week (range: 4–36 sessions for 12 weeks).
Avila et al. (2020) [41]	12 months	Cardiorespiratory fitness Overall VO_2Peak_ (mL/min/kg) and the maximal test duration remained stable at all study groups. VT1 decreased insignificantly in the IGs but remained stable in the CG. No statistically significant differences in responses between groups (P_interaction_ ≥ 0.05 for all).	Physical Activity Decrease in patients with moderate PA > 150′ (*p* = 0.1). No group differences (P_group_ = 0.12). Lower time spent in moderate to vigorous PA (P_time_ = 0.01). Similar in all groups (P_interaction_ = 0.95). Strength/Endurance Improvement in isometric quadriceps and handgrip strength (P_time_ ≤ 0.001). No significant differences among groups (P_interaction_ ≥ 0.05). Cardiovascular risk factors Stable SBP (P_time_ = 0.36). Small increase in DBP (P_time_ = 0.05). Tendency towards higher total cholesterol (P_time_ = 0.09) and LDL values (P_time_ = 0.16) in all three groups. Quality of Life High scores maintained. No between groups interaction in the overall scores and subscores (P_interaction_ = 0.70).
Batalik et al. (2020) [40]	12 weeks	Physical Fitness VO_2_p: improved within both groups ROT (D2.5 ± 3.7 mL/kg/min, *p* < 0.001) and ITG (D2.8 ± 4.7 mL/kg/min, *p* < 0.01), No significant difference between groups. pWL: not statistically significant differences in ROT (D16.3 ± 20.1 W, *p* < 0.001) and ITG (D23.3 ± 31.0 W, *p* < 0.001), nor between the groups.	Quality of Life Total QoL improved significantly in both groups (*p* < 0.01). No significant difference between groups. Adherence ROT patients attended 30.1 ± 6.7 training units (83.6% of all sessions). ITG performed 31.7 ± 8.9 training units (88.2% of all sessions).
Bravo-Escobar et al. (2017) [38]	2 months	Physical Fitness Improved in hospital-based CR group and home-based CR group: METS (*p* = 0.03), recovery rate 1 min (bpm) (*p* = 0.008), exercise time (*p* = 0.03). No between-group differences.	Quality of Life Significant higher in hospital-based CR group (10.93 [IC95%: 17.251, 3.334, *p* = 0.007]). No changes in the home-based CR group (−4.314 [IC95%: −11.414, 2.787; *p* = 0.206]). Adverse events No serious cardiovascular complications or need of hospital treatment.
Cai et al. (2021) [30]	12 weeks	Physical Fitness VO_2_peak [mL/(min × kg)]: improved more in IG (9.3 ± 8.0) than in CG (4.9 ± 6.6) (*p* = 0.003).	Physical Activity Improved more in IG (*p* < 0.001). Health Beliefs Improved more in IG (11.1 ± 10.5) than in CG (2.5 ± 15.2) (*p* = 0.002). Exercise Self-Efficacy Improved more in IG (8.3 ± 4.8) than in CG (4.2 ± 5.3) (*p* < 0.001). Adherence IG attended 9.6 ± 3.1 sessions (80.4% ± 26.1%). CG attended 5.0 ± 3.8 sessions (42.0% ± 31.6%).
Dehghani et al. (2019) [43]	8 weeks	Functional Capacity Significantly improved: MET, VO_2_max, total exercise times (*p* < 0.001) and distance traveled during Bruce test (female: *p* < 0.001, male: *p* < 0.05) in the IG (male and female) compared to the CG. No significant intragroup differences in the CG.	
Frederix et al. (2015) [34]	6 months 2 years	Aerobic Capacity VO_2_peak [mL/(min·kg)] Improved significantly in IG from baseline (22.46 ± 0.78) to 24 weeks (24.46 ± 1.00), *p* < 0.01, decreased from 24 weeks (24 ± 8) to 2 years follow up phase (22 ± 6), *p* < 0.001. No changes in the CG after 24 weeks when compared to baseline (*p* = 0.09) and decreased from week 6 (22.86 ± 0.66) to week 24 (22.15 ± 0.77), *p* = 0.02.	Physical Activity No statistically significant changes in total daily steps (*p* = 0.24). Total daily steps positively correlated with VO_2_peak at baseline (ρ = 0.330, *p* = 0.01), 6 weeks (ρ = 0.237, *p* = 0.03), and 24 weeks (ρ = 0.485, *p* < 0.001). IPAQ. Summed leisure VMW increased significantly in the IG (based on Friedman’s test, χ^2^_2_ = 13.7, *p* = 0.01). No changes in the IG (based on Friedman’s test, χ^2^_2_ = 13.7, *p* = 0.01). Significant between-group difference, in favor of the IG (U = 1830, z = 3.336, *p* = 0.01).Total sitting time decreased significantly in the IG (based on Friedman’s test, χ^2^_2_ = 19.9, *p* < 0.001).Cardiovascular Risk Factors Statistically significant increase only in total cholesterol levels in IG and CG. Health-Related Quality of Life IG increased: perceived HRQL(2.52 ± 0.07; based on Friedman’s test, χ^2^_2_ = 15.4, *p* < 0.001), global HRQL (based on Friedman’s test, χ^2^_2_ = 14.0, *p* < 0.001). No changes in the CG.
Hwang et al. (2017) [32]	12 weeks 24 weeks	Aerobic capacity—6MWD. No significant between-group differences.	No significant between-group differences in balance and muscle strength, QoL. Adherence Higher in the telerehabilitation group. Adverse events Minor (angina, diaphoresis, palpitations).
Kraal et al. (2017) [33]	12 weeks 1 year	Physical fitness VO_2_peak, VAT at VO_2_, peak workload and workload/kg: improved in both groups at 12 weeks and 1 year (*p* < 0.01) without significant between-group differences.	Physical Activity No changes at 1 year period (center-based *p* = 0.38, home-based *p* = 0.80). Quality of Life No significant between-group differences at 12 weeks (*p* = 0.79) and at 1 year follow up (*p* = 0.61). Anxiety Decreased at follow-up in both groups (center-based *p* < 0.05, home-based *p* = 0.01). No differences between groups (*p* = 0.73). Depression No differences between or within groups (*p* < 0.01). Adherence CBCR group attended 20.6 ± 4.3 training sessions. HBCR group performed 22.0 ± 6.8 sessions.
Maddison et al. (2019) [31]	12 weeks 24 weeks	Physical Fitness VO_2_ max: at 12 weeks, comparable in both groups and ITG was non inferior to ROT, (AMD) = 0.51 (95%CI−0.97 to 1.98) mL/kg/min, *p* = 0.48).	Physical Activity At 24 weeks, less sedentary time in ITG (AMD = −61.5 (95% CI −117.8 to −5.3) min/day, *p* = 0.03). BMI Smaller waist (AMD = 1.71 (95% CI 0.09 to 3.34)cm, *p* = 0.04) and hip circumferences (AMD = 1.16 (95% CI 0.06 to 2.27) cm, *p* = 0.04) at 12 weeks in ROT. Cost Evaluation Per capita program delivery (NZD 1130/g BP573 vs. NZD 3466/g BP1758) and medication costs (NZD 331/g BP168 vs. NZD 605/g BP307, *p* = 0.02) were lower for ITG. No statistically significant differences in hospital service utilization costs (NZD 3459/g BP1754 vs. NZD 5464/g BP2771, *p* = 0.20).
Piotrowicz et al. (2015) [33]	8 weeks	Physical Fitness VO_2_peak (mL/kg/min): improved in ITG vs. CG (16.1 ± 4.0 vs. 18.4 ± 4.1 *p* = 0.0001). Significant between-group differences ΔVO_2_peak (Δ2.0 ± 2.4 vs. Δ−0.2 ± 2.1, *p* = 0.0004).	Effectiveness of rehabilitation Workload duration (t) in CPET: improved in ITG (471 ± 141 vs. 577 ± 158 (s), *p* < 0.0001). Significant between group differences: Δt (Δ108 ± 108 vs. Δ0.94 ± 109, *p* = 0.0031). 6-MWT: improved in ITG (428 ± 93 vs. 480 ± 87 (m), *p* < 0.0001). Significant between groups differences: Δ6-MWT (Δ53.8 ± 63.9 vs. Δ22.0 ± 68.7, *p* = 0.0483). QoL: improved in ITG (79.0 ± 31.3 vs. 70.8 ± 30.3 (score), *p* = 0.0001).
Skobel et al. (2017) [37]	6 months	Physical Fitness VO_2_peak (mL/min/kg): improved in ITG vs. CG (1.76 ± 4.1 vs. −0.4 ± 2.7).	QoL, BMI, HR rest, laboratory parameters: no statistical significant changes.
Snoek et al. (2021) [36]	6 months	Physical Fitness VO_2_peak (mL/kg^−1^/min^−1^). Increased after 6 months in MCR group (1.6 [95% CI, 0.9 to 2.4] mL/kg^−1^/min^−1^; relative increase of 8.5%) and 12 months (1.2 [95% CI, 0.4 to 2.0] mL/kg^−1^/min^−1^; relative increase of 6.3%. Change in VO_2_peak higher in the MCR vs. CG at 6 months (+1.2 [95% CI, 0.2 to 2.1] mL/kg^−1^/min^−1^) and 12 months (+0.9 [95% CI, 0.05 to 1.8] mL/kg^−1^/min^−1^.	Physical Activity Self-reported PA greater in MCR group vs. CG (mean absolute difference, 0.7 [95% CI, 0.1–1.3]). Cardiovascular biomarkers DBP and HbA1c stable for the MCR group and increased for the CG. Hospitalization Acute (6 of 19 [3%]) or chronic (8 of 19 [42%]) coronary syndrome.
Song et al. (2020) [39]	6 months	Exercise tolerance VO_2_peak: statistically significant main effect of intervention (*p* = 0.007), main effect of time (*p* = 0.033). Statistically significant differences in VO_2peak_ pred% (*p* = 0.034), HRpeak (*p* < 0.001, AT(*p* = 0.027), VE/VCO_2_@AT (*p* = 0.002), VE/VCO_2_ slope (*p* = 0.002), and OUES(*p* = 0.014).	No statistically significant outcomes

AMD: adjusted mean difference; AT: anaerobic threshold; BMI: body mass index; bpm: beats per minute; CBCR: center-based cardiac rehabilitation; CG: control group; CPET: cardiopulmonary exercise test; DBP: diastolic blood pressure; HBCR: home-based cardiac rehabilitation; HR peak: peak heart rate; IG: intervention group; ITG: interventional home-based telerehabilitation group; LDL: low-density lipoprotein; MCR: mobile cardiac rehabilitation; PA: physical activity; pVO2: peak oxygen consumption; pWL: peak work load; QoL: quality of life; OUES, oxygen uptake efficiency slope; ROT: regular outpatient cardiac rehabilitation; 6MWD: 6 min walking distance; VAT: ventilatory anaerobic threshold; VE/VCO_2_@AT: ventilatory equivalent for carbon dioxide at anaerobic threshold; VE/VCO_2_ slope: the relationship between change in VE and VCO_2_ during incremental exercise; VMW: vigorous and/or moderate and/or walking (VMW) activities; VO_2_peak pred%: percentage of predicted peak oxygen uptake.

## Data Availability

The data presented in this study are available on reasonable request from the corresponding author.

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
