# Peer review of "Effectiveness of Home-Based Cardiac Rehabilitation, Using Wearable Sensors, as a Multicomponent, Cutting-Edge Intervention: A Systematic Review and Meta-Analysis"

_jcm, 2022, doi:10.3390/jcm11133772_

Round 1

Reviewer 1 Report

This is a thorough review with a very sound methodology about a very interesting subject.

The review is well written with good English and only minor corrections required (see specific comments at the end).

I just have two specific points to improve:

1) Image quality in the figures needs to be improved throughout most of the figures - authors need to address this before publication

2) Discussion section: the authors have done a very thorough job on the literature review but I was disappointed with the discussion section. The authors stress the novelty of the results first and then begin to summarise the findings - this should be the other way around. The paper would be much improved if the authors could provide a better summary of the outcome of the review in the discussion section. It is likely that your readers are either clinicians or engineers/scientists. Please consider the likely interest of both of these groups and consider restructuring and adding a few paragraphs to improve the discussion section.

Overall this is a good paper and I make these suggestions to better highlight the findings of this review

Specific minor corrections:

* Line 69 - no full stop before reference

* Line 73 - again no full stop before reference - this needs to be corrected throughout the document

* Line 92 - no capital letter for "Data"

* Improve quality of text and image on figure 2 and 3 - currently looks like screen grab - needs to improve the resolution of the text and image

* Line 387 - two full stops - please remove one

* Image quality of figures 4 and 5 needs to be improved

* Discussion - consider writing a concise summary of your findings before stating the novelty of your work. Also consider adding a few paragraphs to address the interests of your two likely reader groups (clinicians and scientists/engineers).

Author Response

Point 1. Line 69 - no full stop before reference

Response: Thank you for your comment.  Full stop was added before reference

Pont 2. Line 73 - again no full stop before reference - this needs to be corrected throughout the document

Response: Thank you for your comment.  Full stops were corrected throughout the document

Point 3. Line 92 - no capital letter for "Data"

Response: Thank you for your comment.  It was corrected.

Point 4. Improve quality of text and image on figure 2 and 3 - currently looks like screen grab - needs to improve the resolution of the text and image

Response: Thank you for your comment.  The quality of text and image on figures 2 and 3 was improved.

Point 5. Line 387 - two full stops - please remove one

Response: Thank you for your comment.  It was corrected.

Point 6.  Image quality of figures 4 and 5 needs to be improved

Response: Thank you for your comment.  The quality of image on figures 4 and 5 was improved.

Point 7. Discussion - consider writing a concise summary of your findings before stating the novelty of your work. Also consider adding a few paragraphs to address the interests of your two likely reader groups (clinicians and scientists/engineers).

Response: Thank you for your comment.  The discussion-section was revised.

Reviewer 2 Report

In their manuscript, Varsamo & co-workers investigate the effectiveness of wearable sensors for home-based cardiac rehabilitation (HBCR) as an alternative to current strategies. To this aim, the authors provide a systematic review and meta-analysis evaluating the CVD patient’s cardiorespiratory fitness (CRF) and health profile in HBCR population as compared to those in patients receiving standard center-based rehabilitation (CBCR) or usual care (UC). Fourteen randomized studies enabling this comparison were selected for analysis. Significant benefits were observed in the investigated parameters were observed in the comparison between HBCR and CBCR, but not in the comparison between HBCR and UC. Based on their finding, the authors conclude that HBCR can be considered as an adjunct or alternative strategy to CBCR.This is an interesting study. The authors should elaborate further on the possible model enabling the use of HBCR in addition to or as a substitute to the current strategy of CBCR.    

Author Response

Point 1. Based on their finding, the authors conclude that HBCR can be considered as an adjunct or alternative strategy to CBCR. This is an interesting study. The authors should elaborate further on the possible model enabling the use of HBCR in addition to or as a substitute to the current strategy of CBCR.   

Response: Thank you for your comment.  The discussion-section was revised.